# Vinoground: Scrutinizing LMMs over Dense Temporal Reasoning with Short Videos

## Abstract

There has been growing sentiment recently that modern large multimodal models (LMMs) have addressed most of the key challenges related to short video comprehension. As a result, both academia and industry are gradually shifting their attention towards the more complex challenges posed by understanding long-form videos. However, is this really the case? Our studies indicate that LMMs still lack many fundamental reasoning capabilities even when dealing with short videos. We introduce Vinoground, a temporal counterfactual LMM evaluation benchmark encompassing 1000 short and natural video-caption pairs. We demonstrate that existing LMMs severely struggle to distinguish temporal differences between different actions and object transformations. For example, the best model GPT-4o only obtains ∼50% on our text and video scores, showing a large gap compared to the human baseline of ∼90%. All open-source multimodal models and CLIP-based models perform much worse, producing mostly random chance performance. Through this work, we shed light onto the fact that temporal reasoning in short videos is a problem yet to be fully solved. We will make our benchmark publicly available.

## 1 Introduction

Large multimodal models (LMMs) have become very competitive in not only image comprehension but also short video comprehension. Proprietary models such as GPT-4o (OpenAI, 2024a) and Gemini-1.5-Pro (Gemini Team, 2024) as well as open-source models like LLaVA-OneVision (Li et al., 2024a) and Qwen2-VL (Wang et al., 2024) demonstrate strong performance in summarizing a short video's contents and answering questions regarding its details. This has led many researchers to believe that short video comprehension has mostly been solved, and consequently, the community's focus has been increasingly trending toward creating models that understand longer-form videos that are 10s of seconds or even minutes long. Our study, however, indicates that existing models are far from being capable of fully understanding short videos that are just a few seconds long, especially when there is dense temporal information.

As demonstrated in Wu (2024) and Mangalam et al. (2023), for many existing video benchmarks like EgoSchema (Mangalam et al., 2023), ActivityNet-QA (Yu et al., 2019), MSVD and MSRVTT (Xu et al., 2017), the performance of most modern LMMs does not vary significantly with number of sampled frames. In fact, it is often the case that an LMM only needs to see a single frame to produce a correct response. This 'single-frame bias' (Lei et al., 2023) reduces the video comprehension problem into the much easier image comprehension problem, essentially discarding the temporal aspect of a video. Researchers have also proposed harder temporal counterfactual benchmarks (Li et al., 2024c; Saravanan et al., 2024; Liu et al., 2024b) in order to better evaluate an LMM's temporal understanding capabilities. Existing counterfactual datasets test a model's ability to distinguish slight changes from a video's original (positive) caption to the new (negative) caption by asking the model to match the video with the correct caption. However, they either do not contain any negative videos corresponding to the negative caption, or simply swap the order of two unrelated videos to form the positive and negative videos, making it easy to distinguish the negative pair from the original positive pair due to the videos' unnaturalness. Hence, these benchmarks may be inflating the performances of modern LMMs in understanding short videos.

In this paper, we introduce Vinoground, a temporal counterfactual LMM evaluation benchmark composed of 1000 short and natural video-caption pairs. Vinoground is a challenging benchmark

Which video best fits the caption: The man moved the lid from the top to the side?

### In the first video, the man is seen moving the lid from the top of the box to the side. In the second video, the man is seen moving the lid again, but this time it appears to be more about adjusting the lid rather than moving it.
### Conclusion: The caption "the man used one hand to move the lid from the top of the box to the side" matches the first video.

Figure 1: GPT-4o answering a video-score question incorrectly. When asked which video matches the caption, which involves identifying the order of the two events mentioned, GPT-4o does not mention anything about the temporal order of events. The erroneous analyses are marked in red. It should also be noted that the analyses for both videos are completely wrong.

aimed to expose the incapabilities of state-of-the-art models in understanding temporal differences between different actions (e.g., "the man eats then watches TV" vs. "the man watches TV then eats") and object transformations (e.g., "water turning into ice" vs. "ice turning into water"). In each pair of captions, the positive and negative are the same in word composition but different in order. Our work is inspired by Winoground (Thrush et al., 2022), a challenging counterfactual benchmark for visio-linguistic compositional reasoning in images. In Winoground, a model must correctly match two images with their corresponding captions, where both captions use the same set of words, but are rearranged to describe each image (e.g., "some plants surrounding a lightbulb" vs. "a lightbulb surrounding some plants"). This evaluates whether a model effectively encodes the text and images, paying attention to their compositional structures, and whether it can integrate and synthesize information across both modalities. Our benchmark's name changes the 'W' to a 'V' for "video", and further employs temporal counterfactuals to emphasize this unique element in video data. We use text score, video score, and group score to evaluate a model's ability to choose the right caption for a video, to choose the right video for a caption, and to match both positive and negative video-caption pairs correctly, respectively. These measure a model's textual, visual, and temporal reasoning capabilities in a balanced manner. Most of our videos are less than 10 seconds long, yet we find a very large performance gap between an average human and today's best models.

In sum, our main findings and contributions are:

- Existing temporal counterfactual benchmarks fail to fully expose the incapability of LMMs in temporal reasoning.
- We introduce Vinoground, the first temporal and natural counterfactual evaluation benchmark for evaluating video understanding models.
- Modern SoTA LMM performance is subpar when it comes to temporal reasoning in short video comprehension tasks; most models perform at random-chance level on video score and even worse on group score, both being significantly lower than text score.
- We categorize our data into 3 major categories, 'object', 'action', and 'viewpoint', as well as 4 minor categories, 'interaction', 'cyclical', 'spatial', and 'contextual', in order to dissect each model's capabilities for each of these categories. We find that existing models are decent at analyzing video frames at coarse-level but tend to miss fine-grained details.
- Short video comprehension is a problem that is far from being solved.

## 2 RELATED WORK

**Counterfactual Reasoning** Counterfactual reasoning (Morgan & Winship, 2015) in the context of computer vision typically involves curating negative images and captions by manipulating the original data and observing how the outcome changes (Hendricks et al., 2018; Yeh et al., 2019; Goyal et al., 2019; Verma et al., 2020; Guo et al., 2023; Zhang et al., 2021; Thrush et al., 2022; Le et al., 2023; Zhang et al., 2024a). The idea is that a model should understand cause and effect and be able to make predictions in unseen situations. For evaluation, curating meaningful and hard negatives is

important. Winoground (Thrush et al., 2022) is a pioneering benchmark for counterfactual reasoning where each data point contains two images and two corresponding captions. Given an image, a vision-language model is asked to find the matching caption from the provided two options, and vice versa. COCO-Counterfactual (Le et al., 2023) explores simple linguistic rules to generate negative captions and uses an image editing model to produce negative images. In this work, we introduce a novel benchmark with counterfactuals that are temporal, an attribute specific to the video modality.

**Single-Frame Bias and Temporal Reasoning**   An important aspect of video data is its temporality, i.e., how events change as time progresses. Modern LMMs sample frames and treat the video as a set of images, both during training and evaluation. Benchmarks such as EgoSchema (Mangalam et al., 2023), MSVD and MSRVTT (Xu et al., 2017) exhibit a 'single-frame bias' (Lei et al., 2023) where only one video frame is needed for a model to predict correctly, as a model's performance does not vary significantly as the number of frames sampled increases (Wu, 2024; Mangalam et al., 2023). To better evaluate a model's temporal understanding capabilities, researchers have developed datasets such as YouCook2 (Zhou et al., 2018), ActivityNet-QA (Yu et al., 2019) and COIN (Lin et al., 2022), which mainly involve procedural activities that often have a specific temporal dependency (e.g., if a video shows a person washing and slicing apples, and then baking an apple pie, a model would easily predict that "bake it to make a pie before washing the apple" is a wrong caption even without looking at the video). In contrast, Vinoground also includes actions that are entirely unrelated, making it more challenging for models to infer answers based solely on textual cues. MVBench (Li et al., 2024b) also includes temporal data that involves 20 different subcategories of temporal reasoning. However, even with this coverage, this benchmark does not contain any negatives like ours, reducing their difficulty since they do not contain any counterfactual examples. The best models can perform at $\sim 65\%$ on average on their benchmark while Vinoground's best results are far worse ($\sim 35\%$ on one of the metrics).

**Temporal Counterfactuals**   Recent benchmarks combine counterfactuals with temporal reasoning. EgoSchema (Mangalam et al., 2023) introduces long-form videos where each video has 1 positive caption and 4 negative captions to choose from, while VITATECS (Li et al., 2024c) introduces temporal counterfactual data where a word or phrase is swapped/replaced from the positive caption to form the negative caption. However, neither has any negative videos and thus do not fully evaluate an LMM's dense temporal reasoning capabilities like we do. VELOCITI (Saravanan et al., 2024) introduces positive/negative videos as a part of their intra-video association benchmark by clipping random portions in the same video, and asking the model to distinguish between the events. These videos, however, are not truly counterfactual pairs as different clips within the same movie are not guaranteed to have a positive-negative relation. TempCompass (Liu et al., 2024b) includes videos that tests a model's ability to differentiate the order of events, but the videos are either concatenations of two completely unrelated videos with drastic frame changes in between the events, or reversed in time and thus impossible to happen in real life, and do not belong to the true data distribution. As we will illustrate in Section 4.4.2, LMMs tend to do much better when it comes to such videos when compared to our benchmark's more natural negative videos.

## 3   VINOGROUND

In this section, we introduce our data curation and categorization process. In order to curate Vinoground's video-caption pairs, we first explain how we generate the required captions in Section 3.1, how we find the corresponding videos in Section 3.2, and finally the details of categorizing the videos in Section 3.3. An illustration of the overall process can be found in Appendix A.

### 3.1   GENERATING COUNTERFACTUAL CAPTIONS

The first step in curating our data is to find counterfactual caption pairs. We want to ensure that the captions we curate are of high-quality and temporal in nature. While human annotation is a possible solution, it is costly and hard to scale up. Instead, we leverage a SoTA LLM, specifically the GPT-4 (OpenAI, 2024b) model, as it is much cheaper, follows the multiple requirements we impose, and guarantees that there are no duplicate candidates. We require our caption pairs to be composed of the exact same words, only permuted into different orders. We also want to avoid candidates that could easily be solved by looking at a single frame of the video such as "a man is waving at a woman"

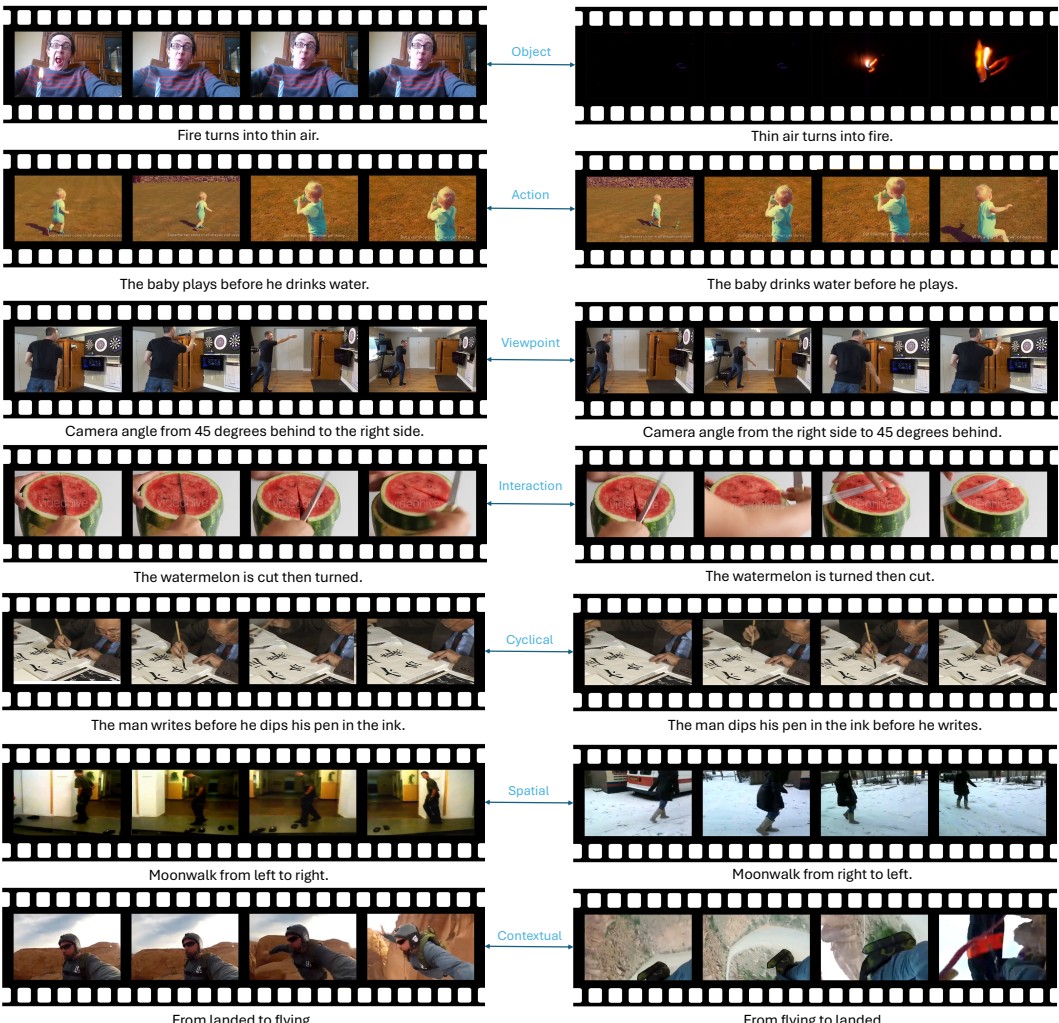

Figure 2: Example positive/negative video-caption pairs in Vinoground, for each category.

vs. "a woman is waving at a man". Hence, we ask GPT-4 to create *temporal* counterfactuals that require one to process and understand the entire video, and in particular, understand the order of events in which they happen, such as "a man waves at a woman before he talks to her" vs. "a man talks to a woman before he waves at her". We will later showcase in Section 4.3 that we can already expose LMMs greatly with such videos (i.e., by swapping the order of two events), making more complicated scenarios unnecessary. We include the detailed prompt we gave to GPT-4 for caption curation in Appendix F.

## 3.2 VIDEO CURATION

After curating counterfactual caption candidates, we next try to find corresponding videos for those captions. We make use of the VATEX (Wang et al., 2019) dataset, which contains 5 distinct captions for each maximum 10-second long video. We only use the validation and test subsets of VATEX to make sure none of Vinoground is ever used as training data. This results in a pool of 9000 videos and 45000 captions.

We want to be able to quickly retrieve potential matches in VATEX according to the generated caption candidates. We leverage sentence transformers (Song et al., 2020), which are good at summarizing sentence-level information into feature vectors, to extract the features of both our GPT-generated captions and VATEX's captions. We subsequently use the Faiss library (Douze et al., 2024) to efficiently index and retrieve the top 20 most similar VATEX captions for each GPT-4 generated caption.

We manually examine if any retrieved caption is a good match, and if its corresponding video reflects the caption as well. For some cases where none of the retrieved captions are a good match, we search YouTube with the caption candidate to find a matching video.

In the end, we curate 500 counterfactual pairs of video-caption pairs (1000 video-caption pairs in total) for evaluation. Each video-caption pair is provided in the form of the original YouTube ID, the clip's starting and ending timestamps, and the corresponding caption. We also put Vinoground through 3 rounds of human evaluation by the authors, making sure that the pair of captions truly contain the same word composition and that the video clips indeed reflect their respective captions.

## 3.3 CATEGORIZATION

Finally, we want to be able to evaluate LMMs in a fine-grained manner on multiple aspects represented by our dataset. Hence, we categorize Vinoground according to the unique characteristics discovered through the data curation process, as shown in Figure 2. We report the number of counterfactual data pairs assigned under each category in Table 1. We define each category as follows:

| Category | Object | Action | Viewpoint | Interaction | Cyclical | Spatial | Contextual |
|---|---|---|---|---|---|---|---|
| Count | 160 | 257 | 83 | 73 | 111 | 103 | 63 |

Table 1: The number of video-caption pairs assigned under each category.

We divide Vinoground into 3 major categories: *object*, *action*, and *viewpoint*. Each counterfactual pair must be in one and only one of the three major categories.

- **Object** requires LMMs to detect changes in the status of one specific object, such as "water turning into ice" vs. "ice turning into water." This category is similar to the "Reversing" category in TempCompass (Liu et al., 2024b) that evaluates a model's ability to detect attribute and directional changes. While TempCompass reverses positive videos in time to create negatives and thus can be unnatural, we curate real, natural videos that correspond to the negative captions.

- **Action**, on the other hand, simply asks models to distinguish the order in which two or more different actions happened, e.g. "the man eats and then watches TV" vs. "the man watches TV and then eats." The two actions need not be correlated at all, and thus less logical comprehension is necessary for a correct prediction.

- **Viewpoint** specifically describes changes in the camera angle, perspective, or focus within the video, such as "a person films the car in front of him before he films himself" vs. "a person films himself before he films the car in front of him." The change in viewpoint is usually accompanied by a drastic difference in between the frames, whereas other events most likely happen within the same context or background.

We also introduce 4 minor categories: *interaction*, *cyclical*, *spatial*, and *contextual*. Some pairs belong to a multitude of these minor categories, while some do not belong to any of them.

- **Interaction** involves videos where a human changes their way of interacting with an object in the course of the video, e.g. "the calligrapher writes with his pen before he dips it into the ink" vs. "the calligrapher dips his pen into the ink before he writes with it."

- **Cyclical** tests a model's ability to identify either procedural temporal activities or two actions that are dependent on each other. The calligrapher example earlier is also cyclical as the person repeats the procedure "write, dip, write, dip...", and the action "dip" happens as a result of "write" in the positive, while "write" is enabled after "dip" in the negative. In contrast, the general "action" category can involve completely unrelated actions.

- **Spatial** It has been shown that LMMs struggle to distinguish physical locations between objects in image-caption pairs (Zhang et al., 2024a). We want to further evaluate this deficiency when it comes to temporal understanding as well. Thus, this category involves object movements and requires positional understanding, such as "the man ran from left to right" vs. "the man ran from right to left." Note that this does not include movement of

the background; e.g., when the camera is moving along with the object in question, which belongs to the next category.

- **Contextual** requires LMMs to understand changes in the background or general information of entire video frames. An example is the pair "the biker rides down the street before he goes down the stairs" vs. "the biker goes down the stairs before he rides down the street" where the camera that records the videos is strapped on the biker's forehead, making the background the only changing aspect. One cannot infer positional changes by only observing movements of the object in the video like the "spatial" category, but instead must focus on the background as the object in question can appear motionless due to the camera moving along with the object.

We provide in-depth analysis of models' performances on our benchmark based on the above categories in Section 4.4.2.

## 4 EXPERIMENTS

In this section, we evaluate state-of-the-art vision-language models on our benchmark. We first describe the models and evaluation metrics in Section 4.1; then we explain our experimental setup, including prompting methods and human studies, in Section 4.2; we analyze the performances of the models in Section 4.3, and provide further ablation studies in Section 4.4.

### 4.1 MODELS AND EVALUATION METRICS

We evaluate both CLIP-based models (Radford et al., 2021) and large generative models, both proprietary and open-source. The exact list of models we evaluate can be found in Table 2. CLIP-based models use contrastive learning between videos and captions, while text-generation LMM models use next-word prediction to generate a response. Due to the different nature of the CLIP-based vs. LMM methods, we introduce our metrics in different fashions accordingly.

We use $C$ to denote captions and $V$ to denote videos. For each positive and negative set of counterfactual video-caption pairs, $(C_i, V_i)$ and $(C_i', V_i')$, $\forall i \in \{1, 2, ..., 500\}$, we ask CLIP-based models to compute a similarity score $e$ between not only the correct pairs but also the incorrect pairs $(C_i, V_i')$ and $(C_i', V_i)$ (identical to Winoground (Thrush et al., 2022)). For generative LMMs, we can only provide inputs (e.g., 2 captions and 1 video) to the model and ask it to choose between the captions/videos.

We first evaluate the text score $s_t$ where the model is presented with both positive and negative captions but only one of the videos, forming the triplets $(C_i, C_i', V_i)$ and $(C_i, C_i', V_i')$. For each triplet, the model is then asked to choose the caption that describes the contained video. We denote the score function of a model response given any triplet as $s$; for instance,

$$s(C_i, C_i', V_i) = \begin{cases} 1 \text{ if LMM chooses } C_i \text{ or } e_{(C_i, V_i)} > e_{(C_i', V_i)} \text{ for CLIP-based} \\ 0 \text{ otherwise} \end{cases}$$

$$s(C_i, C_i', V_i') = \begin{cases} 1 \text{ if LMM chooses } C_i' \text{ or } e_{(C_i', V_i')} > e_{(C_i, V_i')} \text{ for CLIP-based} \\ 0 \text{ otherwise} \end{cases}$$

Then the text score for the given counterfactual pair $(C_i, V_i)$ and $(C_i', V_i')$ is:

$$s_t(C_i, C_i', V_i, V_i') = s(C_i, C_i', V_i) \wedge s(C_i, C_i', V_i')$$

where $\wedge$ is the logical and operator; i.e., $s_t$ is 1 only if both triplets are correct. This exposes the models when they guess randomly.

Similarly, for video score $s_v$, the model is presented with one caption and both positive and negative videos, forming triplets $(C_i, V_i, V_i')$ and $(C_i', V_i, V_i')$. For each triplet, the model is asked to choose the video that is described by the caption. In this case, the response scoring becomes:

$$s(C_i, V_i, V_i') = \begin{cases} 1 \text{ if LMM chooses } V_i \text{ or } e_{(C_i, V_i)} > e_{(C_i, V_i')} \text{ for CLIP-based} \\ 0 \text{ otherwise} \end{cases}$$

$$s(C_i', V_i, V_i') = \begin{cases} 1 \text{ if LMM chooses } V_i' \text{ or } e_{(C_i', V_i')} > e_{(C_i, V_i')} \text{ for CLIP-based} \\ 0 \text{ otherwise} \end{cases}$$

Then the video score is:

$$s_v(C_i, C_i', V_i, V_i') = s(C_i, V_i, V_i') \wedge s(C_i', V_i, V_i')$$

We also include a group score metric $s_g$:

$$s_g(C_i, C_i', V_i, V_i') = s_t(C_i, C_i', V_i, V_i') \wedge s_v(C_i, C_i', V_i, V_i')$$

$s_g$ serves as the ultimate test for a model to demonstrate its temporal reasoning capabilities in both the textual and visual domains, as both $s_t$ and $s_v$ must be 1. For all three metrics, we report the mean over all test instances. We include an illustration of the metrics in Appendix B.

## 4.2 EXPERIMENTAL SETUP

Since for each pair of counterfactuals, we have 2 text-score questions and 2 video-score questions, we have 2000 questions in total. To evaluate CLIP-based models, we use the evaluation code provided by the authors to calculate video-caption embeddings and similarity scores. Evaluating text-generative models is slightly more complicated. We first introduce the different prompts we used. For text score, we provide the model with the video and the two corresponding captions, and prompt "⟨video⟩ Which caption best describes this video? A. {Caption 1}, B. {Caption 2}". For video score, however, since some LMMs only support 1 video input, we concatenate the positive and negative videos into a single video with a 2 second black screen in between. When sampling $N$ frames for the model's input, we make sure we sample $(N-1)/2$ frames from the positive and negative video fragments and at least 1 frame of black screen in between. More details can be seen in Appendix I. For the sake of consistency, we provide all models with the single concatenated video, regardless of how many videos they can actually take as input. We then prompt the model with "⟨video⟩ Which video segment matches this caption? Note: The video contains two segments separated by a 2-second black frame. Caption: {Caption}. A. First segment (before black frame), B. Second segment (after black frame)" to choose between the two video segments. We also report the results with respect to the number of frames sampled by the model from the video, if supported, to evaluate the effect of temporality in Section 4.4.1.

In addition, we also use Prolific (https://www.prolific.com) to evaluate human performance, and find that our dataset is fairly easy for an average human to complete with high accuracy. Prolific is a platform similar to Amazon MTurk which recruits workers to complete tasks such as data annotation. The interface we present to the workers is in Appendix D. To filter out unfaithful workers, we employ a qualification process prior to evaluating on Vinoground. We sample 10 video-question pairs from TempCompass (Liu et al., 2024b) that are of the event order category, which contains concatenated videos with no correlation, such as "a man lifts weights in a gym, then a cat plays on the grass". Such examples are easy enough for an average human to obtain 100% accuracy. We ask the workers the 10 beginner-level questions first, and they are qualified only if they answer every question correctly. This process results in 170 qualified workers, whose demographics are also included in Appendix D.

We conduct human evaluation under two settings. First, the Prolific workers are provided the full videos with audio. To create another environment where we want the workers see the same input as the models, we uniformly sample 32 frames from each video and concatenate them together into a new 10-second video with no audio. The results for the two settings are also compared in Section 4.4.1. For each question, we obtain answers from 10 unique workers. For the 10 answers from a single question, we calculate the *average* human response by taking the mode of the 10 answers. We then report the mean over all the questions as the final result.

## 4.3 MAIN RESULTS

Table 2 presents the results. (Please refer to Appendix H for more detailed results, as we only include each model's best performances here.)

First, all CLIP-based models (VideoCLIP, LanguageBind, ImageBind) perform much worse than random chance, suggesting that contrastive learning does not provide models with enough knowledge of temporality. Among text-generative models, GPT-4o performs best, achieving 54.0% on

| Model | Frames | Text | Video | Group |
|---|---|---|---|---|
| Random Chance | N/A | 25.00 | 25.00 | 16.67 |
| Prolific Human | All | *93.40* | *94.00* | *90.00* |
|  | 32 | *91.40* | *90.80* | *85.20* |
| GPT-4o (OpenAI, 2024a) (CoT) (Wei et al., 2022) | 32 | **59.20** | **51.00** | **35.00** |
| GPT-4o | 32 | **54.00** | **38.20** | **24.60** |
|  | 0 | 10.00 | 24.60 | 2.00 |
| Gemini-1.5-Pro (Gemini Team, 2024) (CoT) | 1fps | **37.00** | 27.60 | 12.40 |
| Gemini-1.5-Pro | 1fps | **35.80** | 22.60 | 10.20 |
| Claude 3.5 Sonnet (Anthropic, 2024) | 4 | **32.80** | 28.80 | 10.60 |
| Qwen2-VL-72B (Wang et al., 2024) | 32 | **50.40** | **32.60** | 17.40 |
| Qwen2-VL-7B (Wang et al., 2024) | 4fps | **40.20** | **32.40** | 15.20 |
| LLaVA-OneVision-Qwen2-72B (Li et al., 2024a) | 32 | **48.40** | **35.20** | **21.80** |
| LLaVA-OneVision-Qwen2-7B (Li et al., 2024a) | 16 | **41.60** | **29.40** | 14.60 |
| InternLM-XC-2.5 (Zhang et al., 2024b) (CoT) | 32/1fps | **30.80** | 28.40 | 9.00 |
| InternLM-XC-2.5 | 32/1fps | 28.80 | 27.80 | 9.60 |
| VideoLLaMA2-72B (Cheng et al., 2024) | 8 | **36.20** | 21.60 | 8.40 |
| MiniCPM-2.6 (Yao et al., 2024) | 16 | **32.60** | **29.20** | 11.20 |
| LLaVA-NeXT-Video-34B (Liu et al., 2024a) (CoT) | 32 | 25.80 | 22.20 | 5.20 |
| LLaVA-NeXT-Video-34B | 32 | 23.00 | 21.20 | 3.80 |
| LLaVA-NeXT-Video-7B (Liu et al., 2024a) (CoT) | 32 | 21.80 | 26.20 | 6.80 |
| LLaVA-NeXT-Video-7B | 32 | 21.80 | 25.60 | 6.20 |
| Matryoshka Multimodal Models ($M^3$) (Cai et al., 2024) | 6 | 21.20 | 25.80 | 6.80 |
| Video-LLaVA-7B (Lin et al., 2024) | 8 | 24.80 | 25.80 | 6.60 |
| Phi-3.5-Vision (Microsoft, 2024) | 16 | 24.00 | 22.40 | 6.20 |
| MA-LMM-Vicuna-7B (He et al., 2024) | 4 | 23.80 | 25.60 | 6.80 |
| VTimeLLM (Huang et al., 2024) | 100 | 19.40 | 27.00 | 5.20 |
| VideoCLIP (Xu et al., 2021) | 60 | 17.00 | 2.80 | 1.20 |
| LanguageBind (Zhu et al., 2024) | 8 | 10.60 | 5.00 | 1.20 |
| ImageBind (Girdhar et al., 2023) | 20 | 9.40 | 3.40 | 0.60 |

Table 2: Vinoground results for different models and sampled frames. Performances significantly better than random chance are bolded. The table is separated into four groups by double lines: random chance and human performance, proprietary text-generative models, open-source text-generative models, and CLIP-based models from top to bottom. The best performances of proprietary and open-source models are highlighted in red.

the text score metric. Chain-of-Thought (CoT) prompting (Wei et al., 2022) further improves GPT-4o's performance, especially on the video score metric where GPT-4o improves by 12.8% while its group score increases by 10.4%. We include the full CoT prompt and parsing process in Appendix G. Amongst the open-source models, LLaVA-OneVision and Qwen2-VL demonstrate competitive performance compared to proprietary models, especially with Qwen2-VL-72B's 50.4% performance on text score. Using CoT on open-source models, however, helps much less, especially if they are performing at near chance level. All other models perform at or worse than random chance, showing that dense temporal reasoning is still very challenging for LMMs.

Similar to Winoground (Thrush et al., 2022), we find that for models that perform better than chance level, their text score is significantly higher than video score, while group score is the lowest amongst all three. This shows that they are better at identifying textual differences compared to visual/temporal differences. For example, GPT-4o's video score (38.20%) is significantly lower compared to its text score (54.0%). Many open-source models only have non-random outcomes on the text score but equal or lower than random chance on video and group scores. Notably, LLaVA-OneVision-72B is the only open-source model that demonstrates better than chance group score.

The human evaluators perform significantly better than any model, with scores around 90%. This indicates that Vinoground is a benchmark that can be tackled relatively easily within human capacity.

| Model | Frames | Text | Video | Group |
|---|---|---|---|---|
| Prolific Human | All | ***93.40*** | ***94.00*** | ***90.00*** |
| | 32 | ***91.40*** | ***90.80*** | ***85.20*** |
| GPT-4o | 64 | **49.00** | **34.80** | 19.00 |
| | 32 | **54.00** | **38.20** | **24.60** |
| | 8 | **53.60** | **31.40** | **20.60** |
| | 1 | 28.20 | 28.00 | 10.00 |
| LLaVA-OneVision-Qwen2-72B | 64 | **46.20** | **31.80** | 18.60 |
| | 32 | **48.40** | **35.20** | **21.80** |
| | 16 | **47.20** | **33.80** | **20.40** |
| | 8 | **46.80** | **29.80** | 19.00 |
| | 4 | **40.40** | 24.80 | 13.00 |
| | 2 | **33.40** | 25.20 | 10.20 |
| LLaVA-OneVision-Qwen2-7B | 64 | **40.20** | 28.60 | 12.60 |
| | 32 | **42.00** | 28.40 | 12.80 |
| | 16 | **41.60** | **29.40** | **14.60** |
| | 8 | **36.00** | 26.80 | 12.40 |
| | 4 | 29.20 | 28.00 | 10.00 |
| | 2 | 25.80 | 22.60 | 6.80 |

Table 3: Results of the strongest closed-source and open-source models with different frames sampled. Performances significantly higher than random chance are highlighted, while the best overall performance of each model are highlighted in red. More frames do lead to better performance, but too many frames can worsen the results.

When the human evaluators are provided with 32-frame videos, the scores decrease by a few points, but are still much higher than those of any model.

Finally, we also report performance for GPT-4o with 0 frames sampled as a control to test for text bias. For text score, we hypothesize that the model will choose the more likely caption since it cannot see the video, and for the video score, we hypothesize it will choose an answer at random, which is indeed what happens. The lower than chance performance for text score of 10.0% indicates that there is some language bias in GPT4o, where it prefers to select one caption over the other (if it consistently did that for all questions, the text score would be 0). Thus, our balanced way of computing the scores (i.e., both $s(C_i, C_i', V_i)$ and $s(C_i, C_i', V_i')$) prevents a model from doing well only via its language bias. This is in contrast to existing benchmarks like VITATECS (Li et al., 2024c) and EgoSchema (Mangalam et al., 2023) which lack negative videos, and hence enable models to potentially answer a question correctly only based on which caption is more likely.

All in all, even the very best models exhibit subpar performance when it comes to dense temporal reasoning, and this is only using short videos (less than 10 seconds) as well. This strongly indicates that short video comprehension in LMMs is still far from human-level intelligence.

## 4.4 IN-DEPTH ANALYSIS OF PERFORMANCE VARIATIONS

### 4.4.1 FRAMES SAMPLED

Vinoground's temporal understanding requirements can be demonstrated by varying the different number of frames sampled, either from the video entirely, or as measured by frames-per-second (fps). If a dataset suffers from 'single-frame bias', a model would not perform very differently when only 1 or more frames are sampled. The results of the strongest proprietary and open-source models in Table 3 (and additional results in Appendix H) show that the more frames a model takes, the better its performance. This indicates that a model does need the entirety of each video to fully comprehend the task at hand. Interestingly, too many sampled frames, however, can hurt a model's performance; for GPT-4o, its 64-frame variant performs 5% worse on all three metrics compared to its 32-frame variant. We suspect that current models are not good at discarding redundant information and isolating signal from noise when there are too many visual tokens.

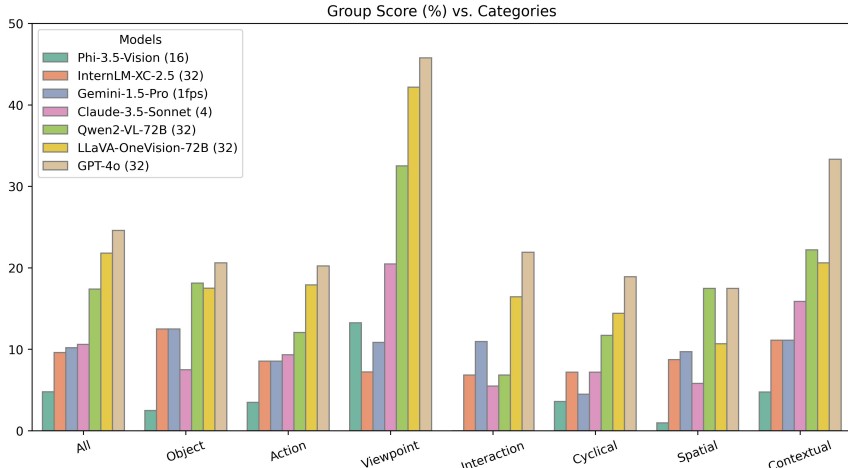

Figure 3: Group score for each model, grouped by category. One can observe higher performance in contextual and viewpoint, and lower performance on other categories.

Note that for our video score metric to function as intended, a model must sample at least one frame from each video, and at least one black frame in between. This means that the number of frames sampled must be no fewer than 3. We hence gray out the video score and group score performances of models sampled at 1 or 2 frames and only focus on their text scores.

Finally, for human evaluators, the 'All' group performs better than the 32 frame group, which indicates that humans can answer Vinoground questions better when the full videos are shown. In contrast, modern LMMs generally lack the ability to process inputs of an entire video without coarse sampling of frames. This suggests that further research into creating models that can handle more frames will be an important research direction for temporal reasoning.

### 4.4.2 CATEGORY

Figure 3 shows results per category as defined in Section 3.3. Interestingly, many models perform significantly better on the *viewpoint* and *contextual* categories, while being significantly worse on other categories. Here, we only report the group score for a selected set of models due to space. Please see Appendix E for the full results.

Both *viewpoint* and *contextual* bring forth drastic changes in between the video frames whenever the events change, as *contextual* involves background changes that occupy most of the frame while in *viewpoint*, as the camera angle changes, the entirety of the video frame changes as well. On the other hand, *interaction* and *cyclical* not only require a model to have strong logical understanding of the connection between events, but also the ability to focus on small temporal changes for the different actions involved. *Spatial*, as previously hypothesized, also poses a difficult challenge for models in understanding changes in object location. Overall, today's models are much better at understanding coarse-level information over a set of frames in their entirety than understanding fine-grained details from a part of each video frame. This also demonstrates how fine-grained comprehension is also crucial for dense temporal reasoning.

## 5 CONCLUSION

We introduced Vinoground, a novel temporal counterfactual benchmark encompassing 1000 short and natural video-caption pairs. We demonstrated that existing video understanding models are quite incapable in terms of temporal reasoning, even for short (<10 seconds) videos. While an average human can easily and accurately complete our benchmark, the best model, GPT-4o, performs much worse, and most models barely perform better than random chance. Our work demonstrates that there is much more to do still in the area of short video comprehension. We believe our benchmark can serve as an important checkpoint in evaluating a model's true performance for temporal understanding of different actions, background transitions, and object transformations.

### LIMITATIONS

One cannot fully analyze the behavior of proprietary models included in this paper due to the lack of access to these models, which are GPT-4o, Gemini-1.5-Pro and Claude 3.5 Sonnet.

## REPRODUCIBILITY STATEMENT

We attach the dataset in the submission's supplementary materials. We will also publicly release it along with the code used to evaluate the LMMs upon the paper's acceptance.

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

APPENDIX

## A  DATA CURATION PROCESS

We include an overall illustration of the data curation process in Figure 4.

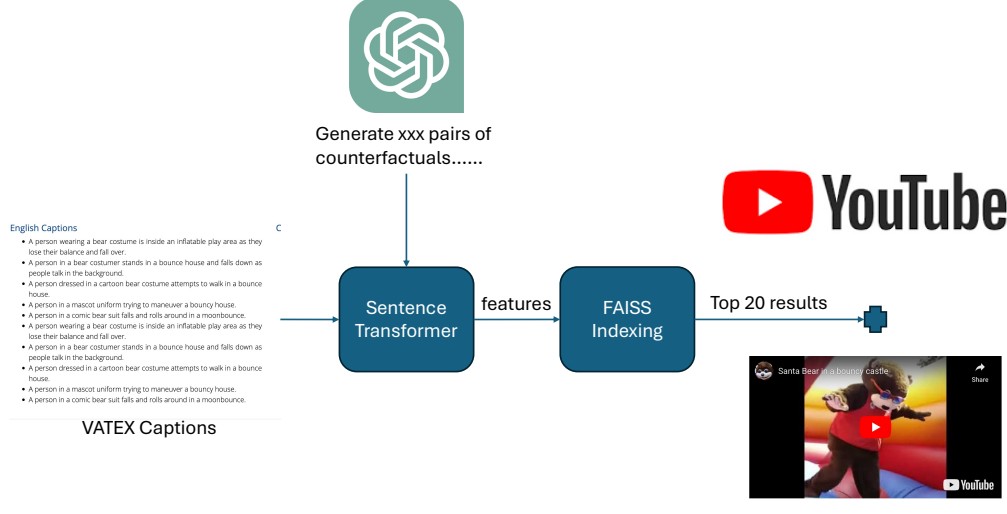

Figure 4: The data curation process.

## B METRICS ILLUSTRATION

We visualize our text and video score metrics in Figure 5. This shows the 4 possible questions that can be derived from one counterfactual data point in the dataset.

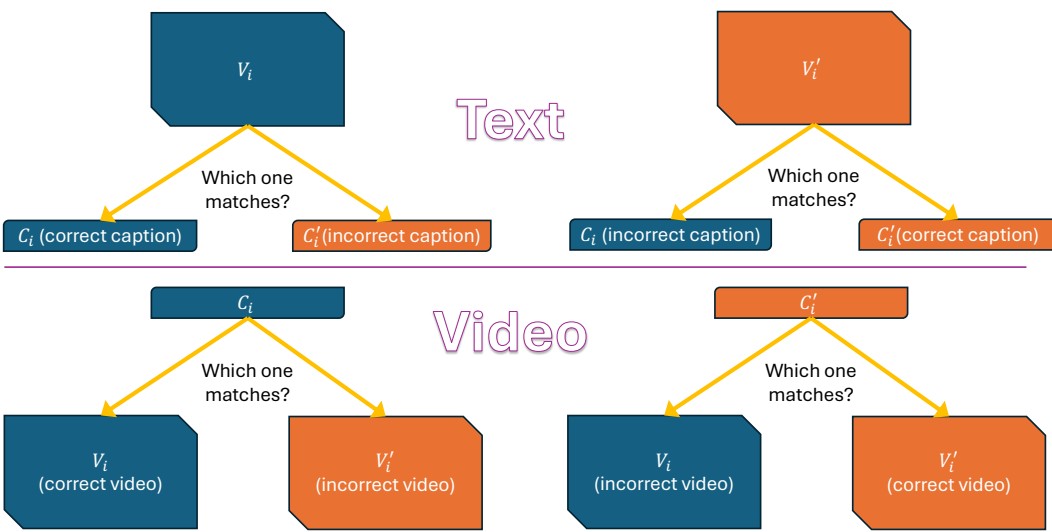

Figure 5: Visualization of the text and video score metrics.

## C RANDOM CHANCE PERFORMANCE

We set the random chance performance for text, video, and group score as 25%, 25%, and 16.67%. It is intuitive to understand the setup for both text and video score since there are two questions in the same counterfactual pair for each metric, and the probability of guessing correctly is 50% each. For the counterfactual pair $(C_i, C_i', V_i, V_i')$, a model can only produce six possible permutations of video-caption matchings: $\{(C_i, V_i), (C_i', V_i')\}$, $\{(C_i, V_i), (C_i, V_i')\}$, $\{(C_i, V_i), (C_i', V_i)\}$, $\{(C_i, V_i'), (C_i', V_i')\}$, $\{(C_i', V_i), (C_i', V_i')\}$, and $\{(C_i', V_i), (C_i, V_i')\}$. This is why the random chance performance for group score is $1/6 = 16.67\%$.

## D PROLIFIC SURVEY INTERFACE AND WORKER DEMOGRAPHICS

We first upload all the videos to Google Drive and embed them into our surveys using Qualtrics. The 2000 questions from Vinoground are split into 50 surveys, with each survey having 40 random questions. We then distribute our surveys on Prolific where we pay everyone who completed a survey $2, or $0.05 per question. The interface is illustrated in Figure 6.

Out of the 170 workers, 91 were male and 79 were female; 13 were students, 119 were non-students, and 38 have no data/did not consent to provide information; 69 were full-time working, 22 were part-time working, 20 were not in paid work (e.g. homemaker, retired or disabled), 17 were unemployed, and 42 have no data/did not consent to provide information; all of the 170 workers were fluent in English; we report the histogram of workers' ages in Figure 7.

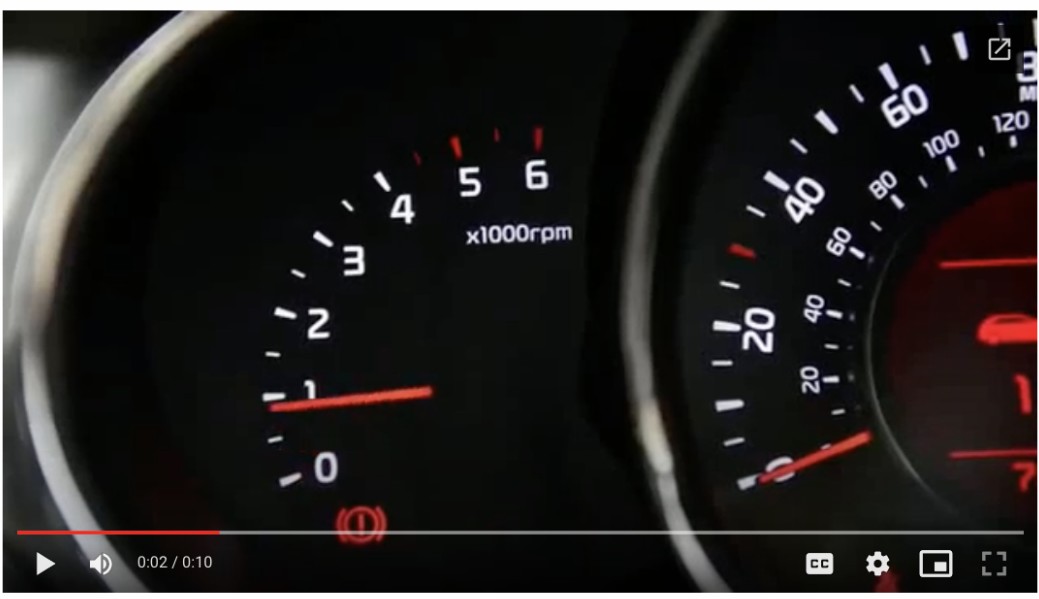

Watch this video, and choose the correct caption out of the two that matches the video.

○ A. the person maneuvers other car parts before he shifts gears

○ B. the person shifts gears before he maneuvers other car parts

Figure 6: The Qualtrics survey that Prolific workers see.

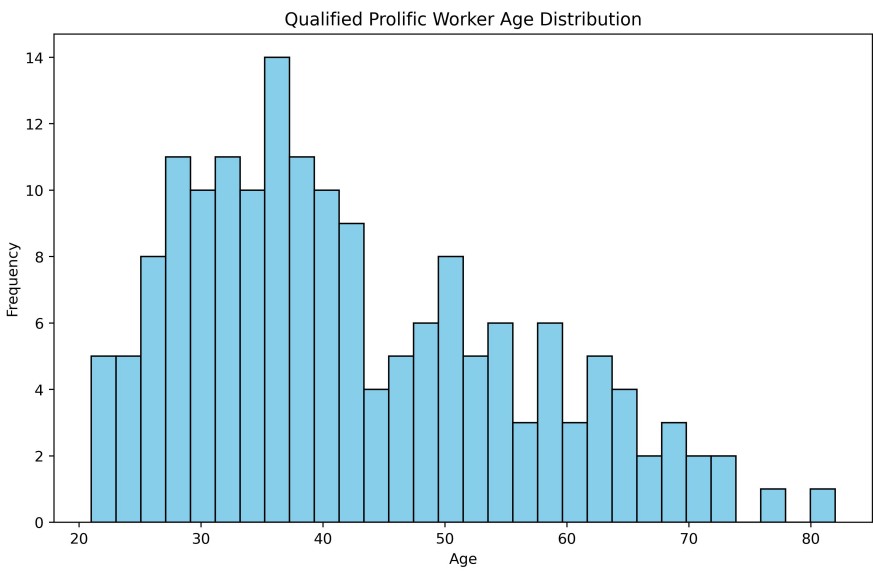

Figure 7: Distribution of Prolific workers' ages.

# E  FULL CATEGORICAL RESULTS

Here we include the selected top-6 strongest models we evaluated and report their results by category in Tables 4 and 5. We also include the text score and video score bar plots in Figures 8 and 9. We can see that the general trend is the same as reported in Section 4.4.2, where models perform much better on contextual and viewpoint, and worse on other categories.

|  | GPT-4o | | | Gemini-1.5-Pro | | | Claude 3.5 Sonnet | | |
|---|---|---|---|---|---|---|---|---|---|
| category | text | video | group | text | video | group | text | video | group |
| all | 54.00 | 38.20 | 24.60 | 35.80 | 22.60 | 10.20 | 32.80 | 28.80 | 10.60 |
| object | 52.50 | 35.62 | 20.62 | 36.25 | 25.62 | 12.50 | 30.00 | 25.00 | 7.50 |
| action | 47.47 | 35.41 | 20.23 | 30.74 | 22.18 | 8.56 | 27.63 | 28.79 | 9.34 |
| viewpoint | 77.11 | 51.81 | 45.78 | 50.60 | 18.07 | 10.84 | 54.22 | 36.14 | 20.48 |
| interaction | 50.68 | 42.47 | 21.92 | 30.14 | 27.40 | 10.96 | 20.55 | 21.92 | 5.48 |
| cyclical | 39.64 | 41.44 | 18.92 | 22.52 | 19.82 | 4.50 | 27.03 | 25.23 | 7.21 |
| spatial | 47.57 | 30.10 | 17.48 | 37.86 | 24.27 | 9.71 | 31.07 | 20.39 | 5.83 |
| contextual | 53.97 | 49.21 | 33.33 | 38.10 | 31.75 | 11.11 | 52.38 | 28.57 | 15.87 |

Table 4: The best performances of proprietary models grouped by category. Significantly high performances are highlighted in blue, while significantly low performances are highlighted in red.

|  | LLaVA-OneVision-72B | | | Qwen2-VL-72B | | | InternLM-XC-2.5 | | |
|---|---|---|---|---|---|---|---|---|---|
| category | text | video | group | text | video | group | text | video | group |
| all | 48.40 | 35.20 | 21.80 | 50.40 | 32.60 | 17.40 | 28.80 | 27.80 | 9.60 |
| object | 42.50 | 33.75 | 17.50 | 46.88 | 33.75 | 18.12 | 28.75 | 28.12 | 12.50 |
| action | 42.80 | 31.91 | 17.90 | 44.75 | 28.79 | 12.06 | 25.68 | 29.96 | 8.56 |
| viewpoint | 77.11 | 48.19 | 42.17 | 74.70 | 42.17 | 32.53 | 38.55 | 20.48 | 7.23 |
| interaction | 36.99 | 36.99 | 16.44 | 34.25 | 31.51 | 6.85 | 23.29 | 36.99 | 6.85 |
| cyclical | 36.04 | 29.73 | 14.41 | 36.94 | 32.43 | 11.71 | 18.92 | 36.04 | 7.21 |
| spatial | 37.86 | 25.24 | 10.68 | 53.40 | 31.07 | 17.48 | 23.30 | 29.13 | 8.74 |
| contextual | 57.14 | 31.75 | 20.63 | 49.21 | 39.68 | 22.22 | 26.98 | 26.98 | 11.11 |

Table 5: The best performances of selected open-source models grouped by category. Significantly high performances are highlighted in blue, while significantly low performances are highlighted in red.

# F  CAPTION CURATION PROMPT

The prompt we gave GPT-4 to generate potential caption candidates is: "I am trying to find videos that have appropriate temporal counterfactuals. e.g., i want to find video pairs that can be described with the following captions: "a man eats then watches TV" vs "a man watches TV then eats"; "the old man is working hard before the young man is playing" vs "the young man is working hard before the old man is playing". Note that for both elements of the same pair, they use the exact same words. give me 10 examples." Then in the same conversation, we prompt the model "give me 10 different ones" until we had 500 pairs of candidates.

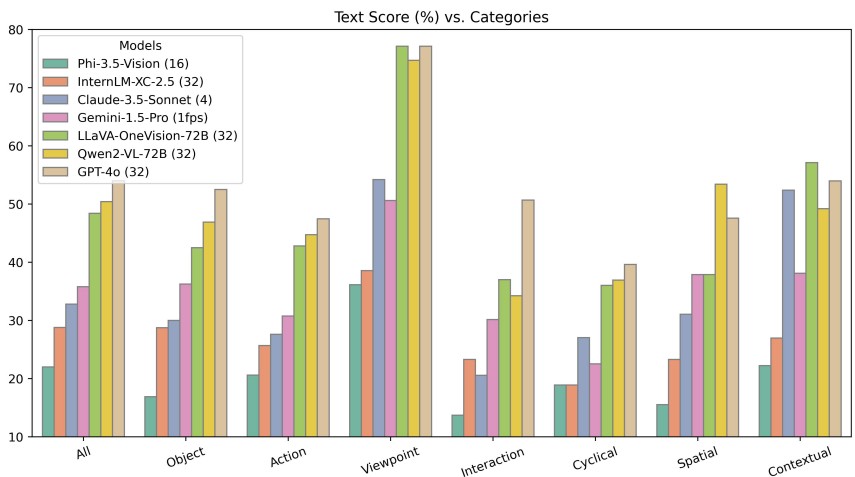

Figure 8: Text score bar plot based on category grouped by model.

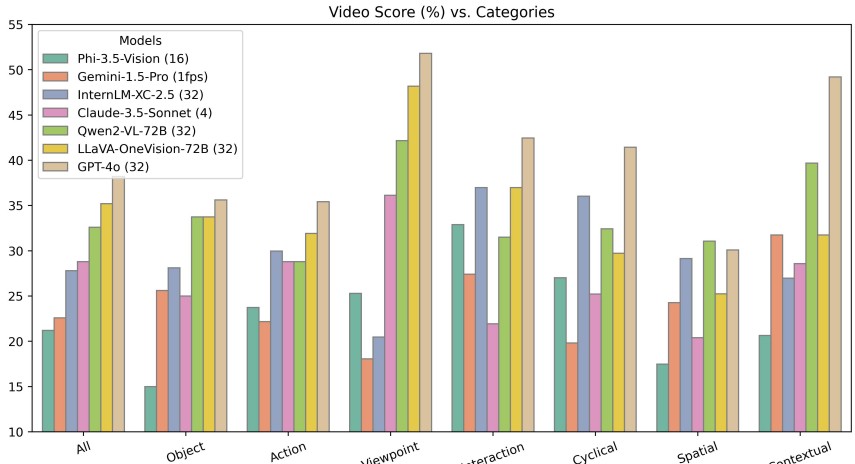

Figure 9: Video score bar plot based on category grouped by model.

## G CoT PROMPT AND PARSING

For chain-of-thought prompting, we simply add "please think step by step" at the end of our questions (as mentioned in Section 4.2). We then use GPT-4 as the judge with the prompt: "Please parse the following model response into either A or B. If the model response is just A or B, then it denotes the model answer, just output it. The model response starts after ====, and end before ====):\n==== ⟨MODEL RESPONSE⟩ ====\nProvide output your answer as a single character (A or B): "

## H FULL RESULTS ON EVALUATED MODELS

Due to the extensive number of models evaluated and different number of samples used as hyperparameters, we include the full results of our evaluation that are not mentioned in the main paper in Table 6.

| Model | Frames | Text | Video | Group |
|---|---|---|---|---|
| Claude-3.5-Sonnet | 16 | **30.00** | 22.60 | 8.40 |
| | 8 | **32.20** | 25.40 | 9.40 |
| | 4 | **32.80** | 28.80 | 10.60 |
| | 2 | 29.40 | 24.00 | 8.40 |
| | 1 | 26.20 | 30.00 | 10.80 |
| Qwen2-VL-72B | 32 | **50.40** | **32.60** | 17.40 |
| | 8 | **37.40** | 23.00 | 7.80 |
| | 4 | 26.20 | 23.80 | 6.20 |
| | 2 | 15.60 | 24.40 | 4.00 |
| Qwen2-VL-7B | 32 | **40.00** | 26.40 | 11.80 |
| | 16 | **36.80** | 25.80 | 10.20 |
| | 8 | 27.60 | 23.40 | 7.80 |
| | 4 | 22.20 | 22.80 | 5.60 |
| | 2 | 21.40 | 25.60 | 5.20 |
| | 4fps | **40.20** | **32.40** | 15.20 |
| | 2fps | **34.80** | 27.40 | 10.60 |
| | 1fps | 26.80 | 26.60 | 7.60 |
| | 0.5fps | 23.20 | 19.60 | 4.80 |
| MiniCPM-2.6 | 32 | 28.40 | 27.00 | 9.40 |
| | 16 | **32.60** | **29.20** | 11.20 |
| | 8 | **33.40** | 25.60 | 9.00 |
| | 4 | 25.80 | 27.40 | 8.60 |
| | 2 | 22.80 | 23.20 | 4.60 |
| | 1 | 27.00 | 27.00 | 8.00 |
| LLaVA-NeXT-Video-34B (CoT) | 32 | 25.80 | 22.20 | 5.20 |
| LLaVA-NeXT-Video-34B | 32 | 23.00 | 21.20 | 3.80 |
| | 16 | 21.00 | 21.80 | 4.40 |
| | 8 | 21.20 | 22.00 | 5.20 |
| | 4 | 16.60 | 21.60 | 3.40 |
| | 2 | 15.40 | 21.60 | 2.20 |
| | 1 | 13.20 | 21.80 | 2.00 |
| LLaVA-NeXT-Video-7B (CoT) | 32 | 21.80 | 26.20 | 6.80 |
| LLaVA-NeXT-Video-7B | 32 | 21.80 | 25.60 | 6.20 |
| | 16 | 22.20 | 25.60 | 6.40 |
| | 8 | 21.80 | 25.60 | 6.40 |
| | 4 | 21.80 | 25.60 | 6.40 |
| | 2 | 21.20 | 25.40 | 6.00 |
| | 1 | 22.40 | 25.60 | 6.40 |
| Phi-3.5-Vision | 32 | 22.00 | 21.20 | 4.80 |
| | 16 | 24.00 | 22.40 | 6.20 |
| | 8 | 21.80 | 21.20 | 5.00 |
| | 4 | 21.20 | 22.80 | 5.60 |
| | 2 | 20.40 | 21.60 | 3.80 |
| | 1 | 22.60 | 22.80 | 3.80 |
| MA-LMM-Vicuna-7B | 32 | 22.40 | 25.60 | 6.80 |
| | 16 | 22.00 | 26.00 | 6.00 |
| | 8 | 23.00 | 26.00 | 6.40 |
| | 4 | 23.80 | 25.60 | 6.80 |
| | 2 | 23.80 | 25.60 | 6.80 |

Table 6: The full evaluation results based on model type, frames sampled, and the metrics aforementioned. Only the model settings that are not mentioned in the main paper are listed here. Performances significantly better than random chance are bolded.

# I  VIDEO LENGTHS AND THE USE OF BLACK FRAMES

We report the video length distribution of our benchmark in Figure 10. We also report that out of the 1000 videos in Vinoground, there are a total of 992 videos with length $\leq 20$ seconds, and 930 of them are $\leq 10$ seconds.

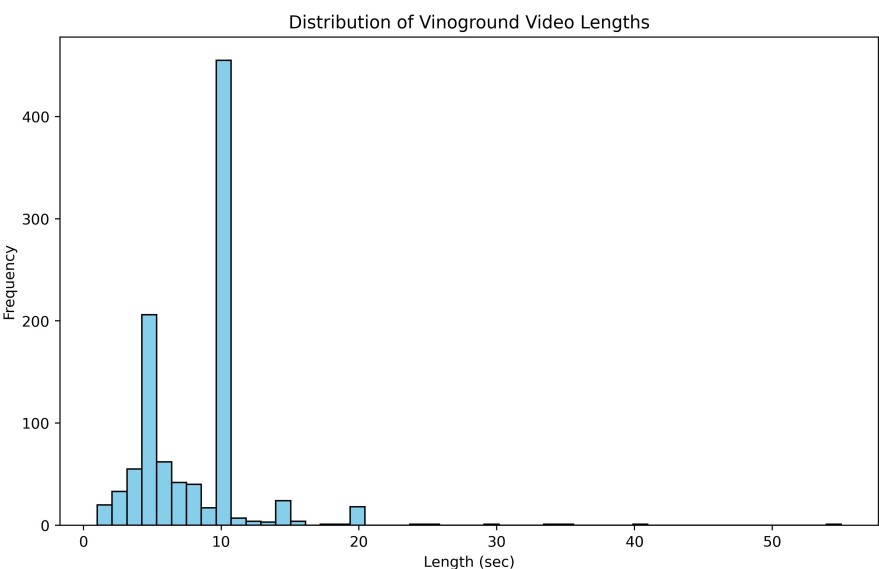

Figure 10: Video length distribution of Vinoground.

Some may argue that there is potential data loss due to concatenation. Here we include here another histogram regarding, in all 500 concatenated videos, how much of each video is composed of black frames in Figure 11.

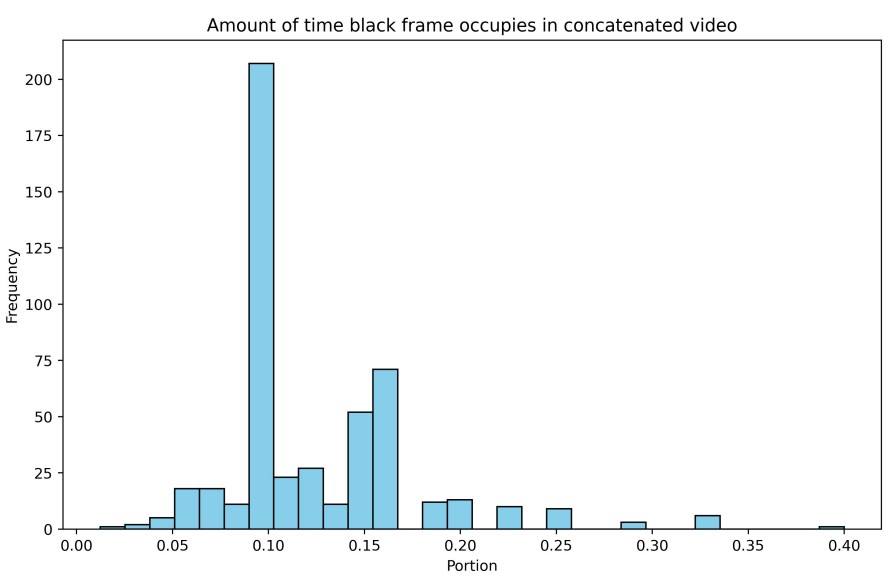

Figure 11: The portion of black frames in each concatenated video for video score questions.

Here we can see that the majority of the videos have only less than one-tenth of the portion containing black frames.

