# OpenReview forum: "Vinoground: Scrutinizing LMMs over Dense Temporal Reasoning with Short Videos"
_ICLR.cc/2025/Conference — Submitted to ICLR 2025_

### Official Review · Reviewer_nPZz · 2024-10-21

**Soundness:** 3
**Presentation:** 2
**Contribution:** 2
**Rating:** 6
**Confidence:** 4

**Summary:**

This paper discusses whether current LLMs truly understand the content of short videos by constructing a temporal counterfactual evaluation benchmark, Vinoground. The benchmark includes 1000 video-caption pairs and assesses the shortcomings of existing LLMs through text, video, and group scores. The evaluation results indicate that dense temporal reasoning within short videos remains a capability that LLMs need to improve and focus on.

**Strengths:**

- The three evaluation metrics proposed in the paper—text, video, and group score—effectively reflect the capabilities of LLM in Vinoground and can expose the model's random guessing behavior.

- This work established a human baseline, providing a good reference for evaluating both open-source and closed-source LLMs.

- This paper reports the performance of GPT-4o under 0-shot sampling as a control for testing bias in these models. This is a highlight.

- The paper presents some new findings, such as the 64-frame variant of GPT-4o performs 5% worse than its 32-frame variant across all three metrics.

**Weaknesses:**

- The author should further elaborate on the principles and methods of generating counterfactual captions, such as how GPT4 is guided to generate counterfactual captions through what kind of prompts. This part of the process is very confusing.

- The author should explain why the generated captions can be found corresponding to videos in the VATEX dataset. I have some doubts about the process of benchmark construction. Shouldn't we first obtain the video and then generate the counterfactual captions?

- The author should provide more detailed classification criteria, such as whether human classification is used, to determine if there is overlap or similarity between action and viewpoint in video-caption pairs.

- Why do some pairs belong to a multitude of these minor categories, while some do not belong to any of them? The minor categories in the benchmark include 350 videos, and I have doubts about the data volume. Due to the imbalance of categories and the reduced number of videos, is the benchmark capable of reasonably evaluating the capabilities of LLMs?

- The author should improve the readability of the article, for example: "For generative LMMs, we can only provide inputs (e.g., 2 captions and 1 video) to the model and ask it to output an answer A or B." Without reading the subsequent experimental setups, the reference to A and B here may lead to confusion.

- The author should provide the video length distribution for each category of the benchmark to illustrate the generalizability of the method where "we concatenate the positive and negative videos into a single video with a 2 second black screen in between." If there are too many short videos, it may result in a significant loss of evaluation data.

- While the article establishes a strong human baseline, the author should provide information on the distribution of factors such as age, gender, educational background, etc., for the human participants. This information would impact the reliability of the human baseline.

- Figures 3, 6, 7, and 8 have low resolution, affecting readability.

I think that there are many unclear points in this paper, especially in the crucial sections on dataset construction and model evaluation, which can lead to misunderstandings.

If my concerns can be addressed, I will raise my rating.

**Questions:**

please refer to "Weaknesses".

---

### Official Review · Reviewer_F3rC · 2024-10-26

**Soundness:** 3
**Presentation:** 3
**Contribution:** 2
**Rating:** 6
**Confidence:** 4

**Summary:**

This paper introduces Vinoground, a benchmark designed to evaluate the temporal reasoning abilities of large multimodal models (LMMs) within the context of short videos. Vinoground comprises counterfactual video-caption pairs, where each pair of captions, generated by GPT-4, contains identical words arranged in different orders. The corresponding videos are collected from two sources: the VATEX dataset and the YouTube platform. The data in Vinoground is systematically categorized into three major categories: *object, action*, and *viewpoint*, along with four fine-grained subcategories: *interaction, cyclical, spatial*, and *contextual*. LMMs are tasked with distinguishing between a pair of counterfactual captions given a video (referred to as text scoring) or between a pair of counterfactual videos given a caption (referred to as video scoring). Empirical results show that current LMMs, including state-of-the-art models like GPT-4, still lag far behind human performance in understanding the temporal dynamics of short videos. They particularly struggle with recognizing fine-grained temporal differences, such as *interaction*, *spatial* (direction), and *cyclical* actions.

**Strengths:**

- Vinoground is specifically designed to expose the true temporal reasoning capabilities of LMMs. By using counterfactual video-caption pairs, it prevents LMMs from relying on single-frame information or language biases, as evidenced by the near random-guessing performance of GPT-4o, when provided with zero or only a single video frame.
- The experiments are comprehensive, encompassing 12 advanced generative LMMs and three CLIP-based models. A detailed analysis is provided on the impact of input frames, the effectiveness of the Chain-of-Thought (CoT) strategy, and the fine-grained performance across different categories.
- The paper is generally well-written and easy to follow.

**Weaknesses:**

- Despite its strengths, the contribution of this work may not be substantial enough for a top-tier conference like ICLR. Compared to existing temporal reasoning benchmarks [1,2,3], the primary innovation of Vinoground lies in the introduction of natural negative videos and a more effective mitigation of single-frame and language biases. While these features do increase the difficulty of the benchmark, current LMMs still show significant room for improvement on existing benchmarks [1,2,3].
- Human performance on the group score is only 90%, indicating potential issues with the quality of the data examples.
- It seems that some critical details are missing when it comes to the generation of counterfactual captions, such as the prompt being used and how many captions are generated.
- A related benchmark [3] is not mentioned or discussed in the paper.

[1] TempCompass: Do video LLMs really understand videos?

[2] Vitatecs: A diagnostic dataset for temporal concept understanding of video-language models.

[3] MVBench: A Comprehensive Multi-modal Video Understanding Benchmark

**Questions:**

N/A

---

### Official Review · Reviewer_LzWR · 2024-10-31

**Soundness:** 3
**Presentation:** 4
**Contribution:** 2
**Rating:** 5
**Confidence:** 4

**Summary:**

This paper introduces Vinoground, a temporal counterfactual benchmark designed to investigate the dense temporal reasoning abilities of video large language models (VidLLMs). The authors propose an elaborate data curation and categorization process to ensure that the counterfactual videos and captions are derived from a natural data distribution, thus guaranteeing data quality. The evaluation results indicate that existing video understanding models struggle significantly with temporal reasoning, even for short videos (less than 10 seconds).

**Strengths:**

1. The paper is well-written, with a clear presentation of the proposed method that is easy to follow.

2. The benchmark design is sound, and the curated data used for evaluation is of high quality. I believe Vinoground will serve as a valuable benchmark for assessing the true performance of VidLLMs in understanding various actions, background transitions, and object transformations.

3. The authors provide a thorough evaluation of current prevalent VidLLMs, revealing their poor performance on dense temporal reasoning tasks, even with short video clips.

**Weaknesses:**

1. **Limited Novelty**:

This is my primary concern. First, the evaluation protocol closely follows that of Winoground. Second, similar concepts for constructing counterfactual caption/video pairs have been proposed in [1][2]. While I acknowledge that the data quality of Vinoground may surpass that of [1][2] due to the curated videos being sourced from natural data distributions, the overall novelty appears to be incremental.

[1] TempCompass: Do video LLMs really understand videos?

[2] Paxion: Patching Action Knowledge in Video-Language Foundation Models.

2. **Lack of Insight into Model Design**:

The evaluation results indicate that existing video understanding models are inadequate in terms of temporal reasoning, even for short videos (less than 10 seconds). However, the authors do not provide in-depth insights into model design. For example, which types of modules could better model temporal information? What training data might enhance performance on temporal reasoning tasks? Such analyses would significantly improve the quality of the paper.

**Questions:**

1. What specific Chain of Thought (COT) prompts were used in the evaluation?

2. Do any time-sensitive VidLLMs (e.g., [3][4]) perform better on Vinoground?

[3] TimeChat: A Time-sensitive Multimodal Large Language Model for Long Video Understanding

[4] VTimeLLM: Empower LLM to Grasp Video Moments

---

> ### Comment · Reviewer_LzWR · 2024-11-23
> **Official Comment by Reviewer LzWR**
>
> Thank you for your responses. I appreciate the evaluation protocol of Winoground/Vinoground and the high data quality of Vinoground. However, I share the concerns raised by Reviewer F3rC regarding the similarity of constructing counterfactual caption/video pairs to previous work. Additionally, the limitations of VideoLLMs in temporal understanding tasks have been well-documented. This work does not appear to offer significant new insights into model design (larger model size/data size leads to better performance is to vague). Accordingly, the contribution of this work may not be substantial enough for a top-tier conference like ICLR, and I will maintain my current rating.

---

### Official Review · Reviewer_ySRv · 2024-11-03

**Soundness:** 3
**Presentation:** 3
**Contribution:** 3
**Rating:** 6
**Confidence:** 3

**Summary:**

This paper presents a dense temporal reasoning benchmark, named Vinoground, for checking CLIP-based and video LLM's capabilities on fundamental temporal counterfactual reasoning tasks. Vinoground contains 1000 video-caption pairs. The authors discuss how they collect counterfactual captions and curate videos for this benchmark. Also, the evaluation can be splitted into three main catergories: object, action, and viewpoint, or four minor categories: interaction, cyclical, spatial, contextual. The authors then discuss. how to evaluate models, either CLIP-based or LLM-based, on this benchmark, and find that today's sota models are struggling to do such temporal counterfactual reasoning tasks, while average crowdsourcing people perform well. The authors present detailed discussions about this results as well.

**Strengths:**

This is overall a good paper investigating the dark side of today's video language models, especially video large language models, on dense temporal counterfactual tasks. The findings are quit interesting: 1. CLIP-based models perform worse than random guessing; 2. Chain-of-thoughts prompting can improve GPT4o but do not help most open-source video LLMs; 3. More frames (larger than 32 for near all models) do not lead to a better result on this dense temporal reasoning task. These may draw to a conclusion: today's video LLMs are performing simple pattern matching  (rather than reasoning). In my view, these findings are valuable and will have impact on further researches on this domain.

**Weaknesses:**

I have no major concerns but it will be more complete that the authors present detailed studies of sota video LLM results on Vinoground, including: 1. on which samples models are prone to fail (hard samples, and their things in common), and easy samples vice versa; 2. why CLIP-based models are strugging with random guessing? Is it due to incapable language encoders? Or incapable visual encoders? 3. Results of temporal random-permuted frames as these results can be more representative than `random chance` in the table since Vinoground is about temporal reasoning.

**Questions:**

See weakness.

---

### Meta-Review · Area_Chair_G8Md · 2024-12-18

**Metareview:**

The paper introduces Vinoground, a benchmark designed to rigorously evaluate the temporal reasoning capabilities of video LLMs using counterfactual video-caption pairs. Its key strengths include the identification of major shortcomings in existing video LLMs, comprehensive experiments across models, novel evaluation metrics (text, video, and group scores), and valuable insights into model behavior under varying conditions, such as frame counts and Chain-of-Thought prompting. However, concerns about limited novelty, benchmark construction transparency, and missing discussions on related work, alongside the absence of deeper analysis into model design and training data improvements, detract from its overall impact. While the findings are significant and the benchmark has potential utility for future research, incremental innovation and gaps in methodological clarity place the paper marginally below the acceptance threshold.

**Additional Comments On Reviewer Discussion:**

The concerns include the similarity of constructing counterfactual caption/video pairs to previous work, and existing works on VideoLLM's limitation in temporal understanding. While most concerns were addressed by the authors (with multiple rounds of discussions needed), the reviewers still share concerns on marginal novelty and insights into model design (larger model size/data size leads to better performance is too vague). This is really a borderline paper.

---

### Decision · Program_Chairs · 2025-01-22

Reject